# Papillary-Muscle-Derived Radiomic Features for Hypertrophic Cardiomyopathy versus Hypertensive Heart Disease Classification

**DOI:** 10.3390/diagnostics13091544

**Published:** 2023-04-25

**Authors:** Qiming Liu, Qifan Lu, Yezi Chai, Zhengyu Tao, Qizhen Wu, Meng Jiang, Jun Pu

**Affiliations:** Department of Cardiology, RenJi Hospital, Shanghai Jiao Tong University School of Medicine, Shanghai 200120, China or 090503liu@sjtu.edu.cn (Q.L.); czzx_2012_505@126.com (Q.L.); cyz960707@126.com (Y.C.);

**Keywords:** radiomics, papillary muscle, hypertrophic cardiomyopathy, hypertensive heart disease, cardiac magnetic resonance

## Abstract

**Purpose**: This study aimed to assess the value of radiomic features derived from the myocardium (MYO) and papillary muscle (PM) for left ventricular hypertrophy (LVH) detection and hypertrophic cardiomyopathy (HCM) versus hypertensive heart disease (HHD) differentiation. **Methods**: There were 345 subjects who underwent cardiovascular magnetic resonance (CMR) examinations that were analyzed. After quality control and manual segmentation, the 3D radiomic features were extracted from the MYO and PM. The data were randomly split into training (70%) and testing (30%) datasets. Feature selection was performed on the training dataset. Five machine learning models were evaluated using the MYO, PM, and MYO+PM features in the detection and differentiation tasks. The optimal differentiation model was further evaluated using CMR parameters and combined features. **Results**: Six features were selected for the MYO, PM, and MYO+PM groups. The support vector machine models performed best in both the detection and differentiation tasks. For LVH detection, the highest area under the curve (AUC) was 0.966 in the MYO group. For HCM vs. HHD differentiation, the best AUC was 0.935 in the MYO+PM group. Comparing the radiomics models to the CMR parameter models for the differentiation tasks, the radiomics models achieved significantly improved the performance (*p* = 0.002). **Conclusions**: The radiomics model with the MYO+PM features showed similar performance to the models developed from the MYO features in the detection task, but outperformed the models developed from the MYO or PM features in the differentiation task. In addition, the radiomic models performed better than the CMR parameters’ models.

## 1. Introduction

Among the various causes of left ventricular hypertrophy (LVH), hypertrophic cardiomyopathy (HCM) and hypertensive heart disease (HHD) are the most-common. However, the detection of LVH and the differentiation between HCM and HHD are often clinically challenging, especially when the hypertrophy is mild to moderate [1]. Meanwhile, the prognosis of patients with HCM varies. In severe and untreated HCM patients, complications include heart failure or even sudden cardiac death [2,3]. For HHD patients, LVH is typically observed in those with long-term poorly controlled hypertension, and without proper treatment, this can progressively develop into heart failure [4]. More importantly, the treatments for these two diseases are quite different [5]. Therefore, based on accurate LVH detection, the early and precise differential diagnosis between HCM and HHD is crucial for treatment plan decisions and prognosis improvement [2].

Compared to other imaging modalities, cardiac magnetic resonance (CMR) offers a noninvasive and multiparametric approach to evaluating cardiac morphology, function, and tissue characteristics [6,7,8]. Its high-resolution cine images also benefit the analysis of both the myocardium (MYO) and small myocardial structures, such as the papillary muscle (PM) [9]. Although small, the PM plays an important role in the maintenance of the mitral valve and left ventricle function [10,11]. Till now, several CMR studies have stressed the importance of the PM on LVH-related diseases: Kozor et al. demonstrated disproportionate hypertrophy of the PM in Fabry disease (FD) and HCM. They also found that the PM mass contributes 8%, 7%, 10%, and 13% to the total weight in healthy subjects, hypertensive patients, HCM patients, and FD patients, respectively [12]. Hoigne et al. confirmed that an abnormal PM was the only significant parameter to diagnose non-obstructive HCM [13]. Despite the morphological differences in the PM in various forms of LVH, the PM’s characteristics have been proven to provide prognostic prediction value for patients with HCM based on image complexity [14].

Despite the direct visual inspection and parameter calculation of CMR images, radiomics provides a “microscopic” approach to analyze CMR morphology and texture characteristics [15]. Radiomics has been developed and researched for over a decade, and its practical applications have been evaluated in many CMR studies [16,17]. We noticed that most studies focused on the MYO features; only recently have some studies extracted the LV features for disease classification [16,18,19]. However, no radiomics studies have attempted to extract the PM features.

According to our clinical experience and observation of CMR images, the PM showed different morphologies, distribution patterns, and signal intensities between patients with LVH and healthy controls (HCs). In this study, we aimed to determine the value of the PM-derived radiomic features in subjects with LVH and HC. Specifically, we sought to develop LVH detection and HCM vs. HHD differentiation models in addition to the PM-derived features.

## 2. Materials and Methods

Figure 1 presents the flowchart of the study.

### 2.1. Study Population

In this retrospective study, patients with LVH were consecutively enrolled in Renji hospitals from August 2018 to December 2021, and HC subjects were randomly selected from our database.

The HCM group inclusion criteria were as follows: (1) genetic determination of an HCM mutation; (2) LVH (end-diastolic wall thickness > 15 mm) in the absence of a clinical condition known to cause hypertrophy; (3) hypertrophy in a recognizable pattern, i.e., apical-variant HCM (AHCM) [2].

The inclusion criteria for the HHD group were as follows: (1) echocardiogram (ECG) demonstration of a hypertrophic LV (maximal LV wall thickness > 11 mm or LV-mass-to-body-surface area > 115 g/m${}^{2}$ for men or > 95 g/m${}^{2}$ for women) in the absence of other cardiac or systemic diseases) [20]; (2) diagnosis of arterial hypertension [21].

The HC group consisted of healthy volunteers who demonstrated normal cardiac dimensions and volumes, normal cardiac function, and the absence of late gadolinium enhancement. None of the control subjects had a history of known cardiac disease, including cardiac surgery or interventions.

Exclusion criteria for all subjects were an established diagnosis of FD, cardiac amyloidosis, severe valvular disease, aortic stenosis, iron deposition, evidence of inflammatory processes in the myocardium or pericardium, history of ST segment elevation myocardial infarction, and LVH caused by athlete’s heart [22].

### 2.2. Image Acquisition and Analysis

CMR examinations were performed in Renji hospital with a 3T MR scanner (Ingenia, Philips). Cine images were acquired using a balanced steady-state free-precision (bSSFP) sequence [23]. Other typical cine parameters were: slice thickness = 6–8 mm, echo time (TE) = 1.4 ms, repetition time (TR) = 3.0 ms, flip angle = 45°, in-plane resolution = 0.8–1.2 mm × 0.8–1.2 mm, bandwidth = 1900 Hz/pixel, number of cardiac phases = 30. For all participants, the cine stack was covered from the base to the apex.

The CMR data were exported in the Digital Imaging and Communications in Medicine (DICOM) format. Anonymization and CMR function analyses were performed on a workstation with dedicated post-processing software cvi42 (Version 5.13.0, Albert, Canada). All images were subjected to manual quality control to exclude those of inadequate quality before further assessment. Basic CMR parameters that summarize LV function (including LV mass, LV end-diastolic volume (LVEDV), and LV ejection fraction (LVEF)) were obtained using the cvi42.

### 2.3. Definition of Volume of Interest

Limited by copyright, we were not able to export contours directly from the cvi42 software. To export the MYO and PM contours, an open-source software itk-snap (Version 3.8.0) was used for the following analysis. All images were segmented simultaneously by two cardiologists (>4 years of CMR experience) in consensus. After that, another experienced cardiologist (>5 years of CMR experience) checked the contours and made adjustments if necessary. In this study, only ED cine images were considered for analysis due to ambiguous border between the endocardium and PM in end systole. The VOI of the MYO was defined as the area between the LV endocardium and epicardium [24]. For the VOI of the PM, we delineated the PM contours according to anatomical characteristics, movement pattern, and previous segmentation examples [12,25]. non-PM structures (e.g., papillary muscle variants, apical–basal muscle bundle etc.), and trabeculations were excluded [9,26].

Appendix A shows a representative apical–basal muscle bundle, which is hard to distinguish from the PM with short-axis cine images solely in our datasets.

### 2.4. Feature Extraction and Feature Selection

Image filters (wavelet, Laplace of Gaussian (LOG), and gradient) were applied to our datasets, and 3D radiomic features were extracted for both the MYO and PM VOIs using Pyradiomics [27]. The detailed description and calculation equation are available on the Pyradiomics official website: https://pyradiomics.readthedocs.io/en/v3.0.1/radiomics.html, accessed on 1 October 2022.

After feature extraction, the data were randomly split into training and testing datasets in a 7:3 ratio.

Feature selection was performed on the training dataset with 3 steps: (1) The Pearson correlation coefficient (ρ) was calculated for each feature; features with ρ > 0.8 were excluded; (2) a least absolute shrinkage and selection operator (LASSO) regression algorithm was employed to select important features [28]; (3) features were ranked using the Boruta method [29].

### 2.5. Model Development

Five machine learning classifiers were evaluated: AdaBoost (AB), K-nearest neighbor (KNN), support vector machine (SVM), random forest (RF), and decision tree (DT). A five-fold cross-validation (CV) with a grid search was performed within the training dataset to determine the best combination of model parameters. The performance of the different models was compared based on the area under the curve (AUC), accuracy, precision, recall, and calibration curve.

### 2.6. Evaluation of MYO and PM Features

For the multi-feature analysis, we aimed to assess the value of the PM features. Therefore, three radiomics groups were designed: (1) the MYO group, including the best 2 × N MYO features; (2) the PM group, including the best 2 × N PM features; (3) the MYO+PM group, including the best N MYO and N PM features. Subsequently, different machine learning (ML) models were tested and evaluated in the three groups.

### 2.7. Comparison of Radiomic Models and Clinical Data Models

To compare our proposed radiomics models with clinical data models, an optimal ML model was selected and developed using CMR parameters (LVEF, LVEDV index, and LV mass index). We also combined the CMR parameters and radiomic features to examine the possible synergistic effects between CMR parameters and radiomics information. The development and evaluation processes were the same as those previously described.

### 2.8. Statistics

All statistical analyses were performed using SPSS (Version 26) and Python (Version 3.7). To compare the means, Student’s *T*-test, the paired *T*-test, and the Mann–Whitney U-test were conducted as appropriate. Statistical significance was set at *p* < 0.05. Model performance was evaluated using the receiver operating characteristic (ROC) curves and AUC, accuracy, precision, and recall, and a comparison between the AUCs was performed with the Delong test [30]. The Hosmer–Lemeshow test was used for the calibration curve evaluation [31]. Weights were calculated for the HC, HCM, and HHD groups, where appropriate, to compensate the unbalanced datasets [32].

## 3. Results

### 3.1. Demographic and CMR-Based Clinical Characteristics

The detailed inclusion and exclusion pipeline is shown in Figure 2.

A total of 230 LVH (NHCM =158, NHHD = 72) patients who satisfied the inclusion criteria and passed quality control were included. Subjects in the HC group were matched with patients with LVH in terms of age and sex (*p* = 0.866 and 0.364, respectively) at a 1:2 ratio. Table 1 shows the demographic and CMR data of the cohort.

For the CMR parameters, subjects with LVH showed a significantly increased LV mass (144.3 g vs. 83.2 g, *p* < 0.001) and LV mass index (80.0 g/m${}^{2}$ vs. 46.3 g/m${}^{2}$, *p* < 0.001). However, for the HHD group, we found that their LVEFs were lower than those in the HCM and HC groups (55% vs. 68% and 65%, respectively, *p* < 0.001 for both groups), whereas the LVEDVs were higher than those in the HCM and HC groups (163 mL vs. 130 mL and 126 mL, respectively, *p* < 0.001 for both groups). This phenomenon could be partly explained by the fact that HHD patients usually have had hypertension for many years; therefore, some patients already had an enlarged left ventricle.

### 3.2. Feature Extraction and Selection

A total of 2632 features (N${}_{MYO}$ = 1316 and N${}_{PM}$ = 1316) were extracted; the detailed feature information is available in Appendix A. After correlation and LASSO feature selection, 60 features (N${}_{MYO}$ = 28, N${}_{PM}$ = 32) survived; meanwhile, for the detection task, 20 features (N${}_{MYO}$ = 8, N${}_{PM}$ = 12) survived, and the features were ranked using relative importance calculated from the Boruta method. Table 2 shows the six most-important features for the differentiation tasks derived from the MYO and PM. The gradient gray-level co-occurrence matrix (GLCM) correlation was the most-important in the MYO group, and the shape maximum 2D diameter slice was the most-important in the PM group. Complete lists of the detection and differentiation tasks are provided in Appendix A.

### 3.3. Multi-Feature Analysis of MYO, PM, and MYO+PM Groups

For both the detection and differentiation tasks, we used N = 3 for further analysis; therefore, we had six features in the MYO, PM, and MYO+PM groups. Among the six features in the MYO+PM group, there were three shape features (MYO: shape sphericity and shape elongation; PM: shape maximum 2D diameter slice).

For the detection task, different model performances on the training dataset are shown in Figure 3. After comparing the AUC, accuracy, and calibration curves (Appendix A), the SVM models were selected. The performance of the SVM models with the testing dataset is presented in Table 3. Among the MYO, PM, and MYO+PM groups, the model developed with the MYO features achieved the highest AUC (0.966 vs. 0.944 (MYO+PM) and 0.772 (PM), *p* = 0.924 and *p* < 0.001, respectively); however, accuracy, precision, and recall were slightly higher in the MYO+PM group. The results indicated the MYO features had a similar efficacy as the MYO+PM features in the detection task.

For the differentiation task, the z-score distribution of the MYO+PM group is shown in Figure 4. The performances of the different models on the training dataset are shown in Figure 3, and the calibration curves are shown in Appendix A. Similar to the detection task, SVM was selected for further analysis. The results of the SVM model showed that the MYO+PM group had a significantly higher AUC (0.935 vs. 0.875 (MYO) and 0.716 (PM), *p* = 0.040 and 0.002, respectively), and the accuracy showed a 4.4% increase (87.0% vs. 82.6%). The ROC and calibration curves for the differentiation task with the SVM models are plotted in Figure 5, which shows excellent calibration results (all *p* > 0.05, indicating excellent calibration results) for both the training and testing data. The clinical usefulness of the MYO, PM, and MYO+PM models is shown as decision curves in Figure 6.

### 3.4. Comparison of Radiomics Models to CMR Parameter Models

As shown in Table 4, the radiomics model showed significant improvements compared with the CMR parameter models (AUC: 0.935 vs. 0.774, *p* = 0.002), and no significant improvement was observed when comparing the radiomics + CMR parameter model with the radiomics model (AUC: 0.935 vs. 0.906, *p* = 0.117). We also observed that the CMR parameters group showed biased results (precision = 0.693, recall = 0.409, and F1 = 0.474) in the differentiation task. The calibration curves for the models developed with the CMR parameters, radiomics, and radiomics + CMR parameters are exhibited in Figure 7. The clinical usefulness of the radiomics, CMR, and radiomics+CMR models is shown as the decision curves in Figure 6.

## 4. Discussion

In this study, we investigated the LVH detection and HCM vs. HHD differentiation ability of the MYO, PM, and MYO+PM radiomic features; we also compared the MYO+PM group’s results with CMR parameters group’s results.

### 4.1. Summary of Main Findings

The main findings of our study were as follows:The MYO and MYO+PM groups showed great LVH detection based on the AUC and accuracy;The MYO+PM group outperformed the MYO group on the HCM vs. HHD differentiation task;Our proposed radiomics models showed significantly better performance than the CMR parameter models.Our methods showed excellent calibration results and high clinical usefulness, as shown by the calibration curves and decision curves.

The main findings are summarized in Table 5.

### 4.2. Discussion Based on Results

Radiomics analysis has been widely applied for the differentiation of cardiomyopathies. However, to our knowledge, this is the first study focusing on the radiomics of the PM. Previous radiomics studies on cardiomyopathy classification were mainly based on the MYO features. Ulf et al. extracted radiomic features from native T1 MYO and selected six texture features; their final model achieved a 0.80 accuracy and 0.89 AUC on the test data [24]. Xu et al. combined deep learning (DL) and radiomics methods for ECG-based LVH aetiology differentiation; their final results showed an AUC of 0.839 for HCM vs. HHD classification [33]. Izquierdo et al. compared their radiomics models to CMR-index-based models and found no significant differences [19]. Although the model performance varied with the datasets, in this study, our models showed satisfactory HCM vs. HHD classification results (accuracy = 87.0% vs. 80.0%, AUC = 0.94 vs. 0.89) compared with previous studies. Using the CMR parameter model as the baseline, our radiomics model showed significant improvement (AUC:0.935 vs. 0.774, P = 0.002). Based on these results, we demonstrated the effectiveness and superiority of the proposed radiomics models.

In addition, our models achieved excellent results with limited features (three MYO + three PM features), and the number of features selected was also consistent with several previous studies [24,34]. Among the six features selected, three belonged to the shape feature class (MYO: shape sphericity and shape elongation; PM: shape maximum 2D diameter slice), which made our concise MYO+PM radiomics model more explainable. Radiologically, hypertrophy in HCM patients usually show a concentric pattern, while the hypertrophy of HHD is mainly affected by high blood pressure, which made “MYO sphericity” and “MYO elongation” more reasonable from a clinical perspective. “PM maximum 2D diameter slice” represented the largest distance between the surface mesh vertices in the slice level, which could reflect the different distribution patterns of the PM in HCM and HHD. These results also validated our observation that the PM’s morphology can facilitate the classification between HCM and HHD.

From a clinical point of view, no diagnosis relies purely on radiomic features; therefore, we also developed CMR parameter models and radiomics + CMR combined models. Consistent with a previous study, our CMR-parameter-based model showed unsatisfactory prediction accuracy of only 81.7% and 71.0% on LVH detection and HCM vs. HHD differentiation, respectively [1]. After the incorporation of radiomic features, the performance improved in both tasks, but for HCM vs. HHD, the AUC and accuracy were still worse than the models using solely radiomic features (Table 5).

### 4.3. Technical Perspectives

Despite these promising findings, we considered the possibility of over-fitting. Over-fitting is a common problem in radiomics studies. Therefore, we implemented calibration curves for different ML models to facilitate both model selection (Appendix A) and model performance evaluation (Figure 5 and Figure 7). Although the SVM and KNN methods both showed excellent calibration results, the SVM models showed higher performance than the KNN models in both the detection and differentiation tasks (Figure 4). Therefore, the SVM models were selected for further analysis. The following experimental results validated that the SVM models performed well and did not show an over-fitting tendency with the six radiomic features as the inputs (Figure 5).

From a technical perspective, to deal with unbalanced datasets, class weights were calculated for each group when appropriate. Although we did not compare the performance of our “balanced” dataset with that of an unbalanced dataset, our results showed the following: except for the PM group, the MYO and MYO+PM groups exhibited relatively high precision (all > 0.8) and recall (all > 0.75) on both the training (Figure 3) and testing (Table 3 and Table 4) datasets.

### 4.4. Clinical Perspectives

The implementation of both the MYO and MYO+PM features showed excellent LVH detection performance; however, the MYO+PM group outperformed the MYO group on the HCM vs. HHD differentiation (*p* = 0.040). We also validated that the incorporation of the CMR parameters into the radiomic information did not facilitate HCM vs. HHD differentiation. Our results suggest to clinicians that the PM is a potential useful diagnostic tool for LVH. In addition, we suggest that the calibration curve should be examined for every machine learning algorithm, if appropriate.

### 4.5. Limitations

This study had some limitations.

First is the immature DL-based PM automatic segmentation results of the PM (DSC = 0.79 in previous studies) [25,35]. In this study, we performed manual segmentation and quality control of selected VOIs. To increase the segmentation accuracy and minimize interobserver variance, two cardiologists completed the segmentation together in consensus. Although time consuming, accurate manual segmentation is crucial for study reliability and repeatability. We also look forward to developing a precise and stable DL segmentation model for further implementation of deep PM radiomics studies.

Second, LVH is not a singular disease entity. The different pathologies of LVH exhibit various clinical behaviors, treatment responses, and prognoses. HCM and HHD are two most-common causes of LVH. Therefore, only patients with HCM or HHD were included in this study. If the sample size permits, we would also like to validate our methods using other available LVH-related disease datasets.

## 5. Conclusions

To the best of our knowledge, this is the first study on PM-derived radiomic features. Our results showed that the proposed PM radiomics exhibited excellent performance in LVH detection and HCM vs. HHD differentiation and outperformed clinical data models. Moreover, through our experiments, we hypothesized that PM radiomics have great potential for the classification of other cardiac diseases and could facilitate clinicians’ clinical decision-making.

## Figures and Tables

**Figure 1 diagnostics-13-01544-f001:**
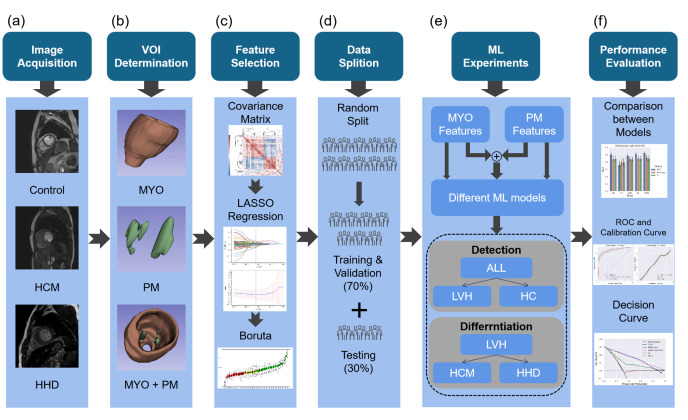
Schematic view of this study. (**a**) CMR data were collected from HCM, HHD, and HC subjects, and typical mid-ventricle cine images are shown; (**b**) examples of the volume of interest (VOI) for the MYO, PM, and MYO+PM together in an HC subject (visualization was performed with a 3D slicer); (**c**) described methods used in the feature selection; (**d**) shows how the data partition was performed; (**e**) described ML pipeline: selected MYO and PM features were evaluated for detection and differentiation performance with different ML methods; (**f**) results were compared between different groups and ML methods, and the evaluation was performed with the ROC curve, calibration curve, and decision curve. HCM, hypertrophic cardiomyopathy; HHD, hypertensive heart disease; VOI, volume of interest; LASSO, least absolute shrinkage and selection operator; ML, machine learning; MYO, myocardium; PM, papillary muscle; LVH, left ventricular hypertrophy.

**Figure 2 diagnostics-13-01544-f002:**
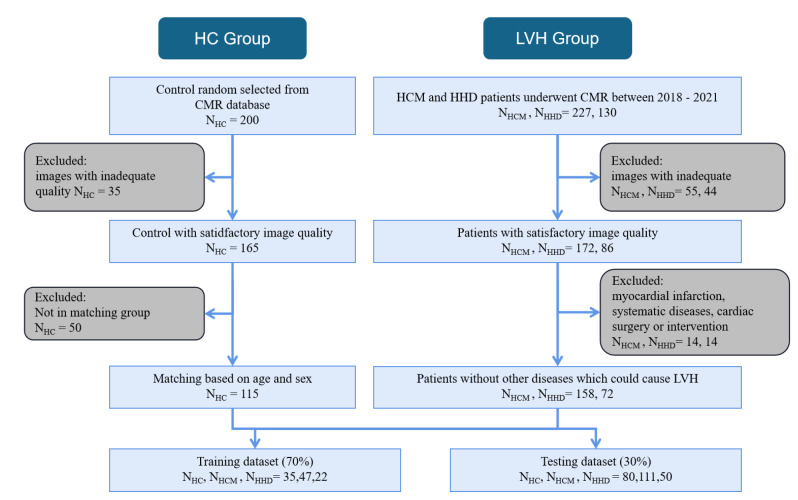
This figure shows the inclusion and exclusion of LVH and HC subjects. All subjects were obtained from our datasets between 2018 and 2021. For the LVH group, firstly, images with inadequate quality were excluded. Secondly, subjects with myocardial infarction, systemic diseases, or cardiac surgery were further excluded. For the HC group, images with satisfactory quality were matched to the LVH group by sex and age at a 1:2 ratio.

**Figure 3 diagnostics-13-01544-f003:**
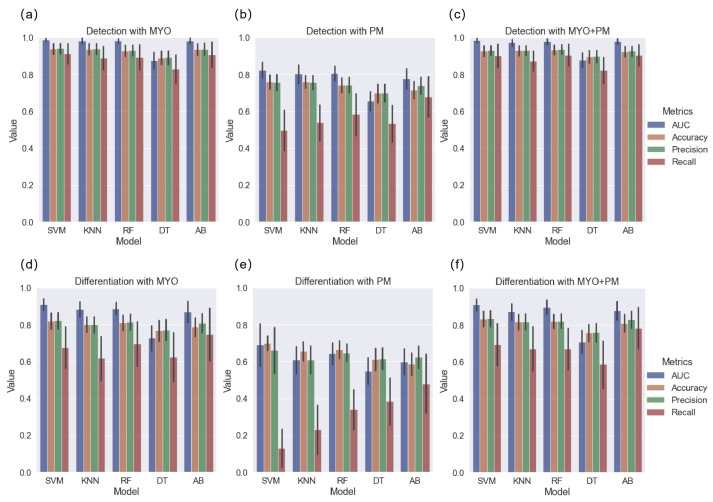
Performance (AUC, accuracy, precision, and recall) of different models on the detection and differentiation tasks with 3 radiomics subgroups; (**a**–**c**) detection task with different ML methods using MYO, PM, and MYO+PM groups, respectively; (**d**–**f**) differentiation task with different ML methods using MYO, PM, and MYO+PM groups, respectively. SVM, support vector machine; KNN, K-nearest neighbor; RF, random forest; DT, decision tree; AB, AdaBoost.

**Figure 4 diagnostics-13-01544-f004:**
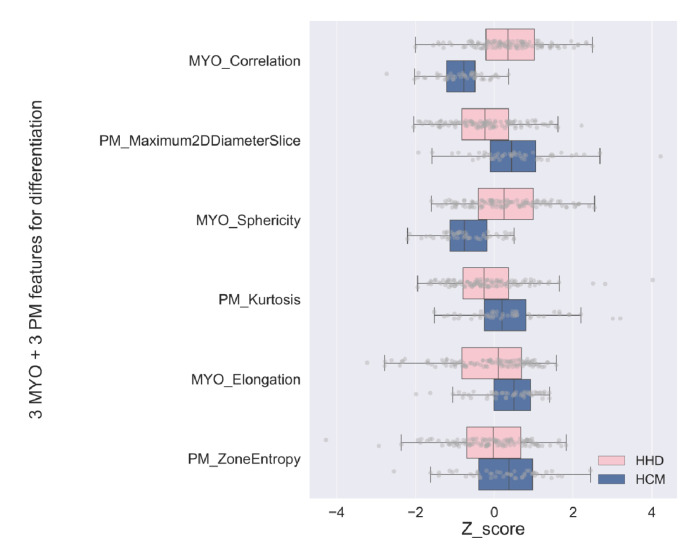
Box plot for the best 3 MYO and 3 PM features in differentiation task. The pink and blue boxes represent the range between the first quartile to the third quartile, while the gray dots represent each single patient in the training dataset. Only features in the MYO+PM group for the differentiation task are exhibited.

**Figure 5 diagnostics-13-01544-f005:**
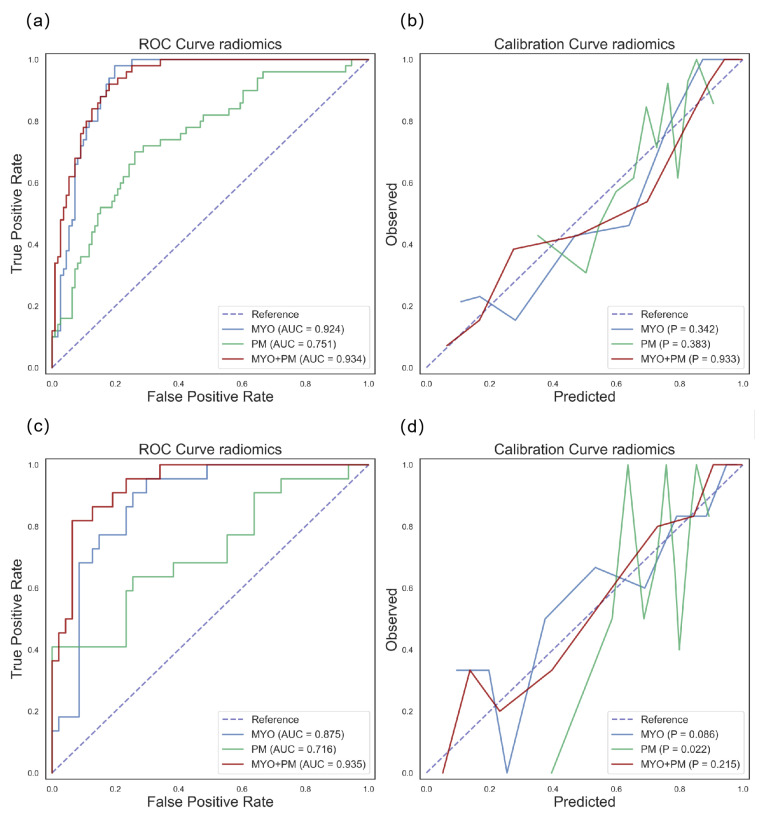
ROC curves and calibration curves for differentiation task with MYO, PM, and MYO+PM groups using SVM models. Blue, green, and red lines represent the MYO, PM, and MYO+PM groups, respectively; the AUC for the ROC curves and Hosmer–Lemeshow *p*-values are denoted in the figure legends. The first row shows the results on the training dataset: (**a**) ROC curve and (**b**) calibration curve. The second row shows the results on the testing dataset: (**c**) ROC curve and (**d**) calibration curve.

**Figure 6 diagnostics-13-01544-f006:**
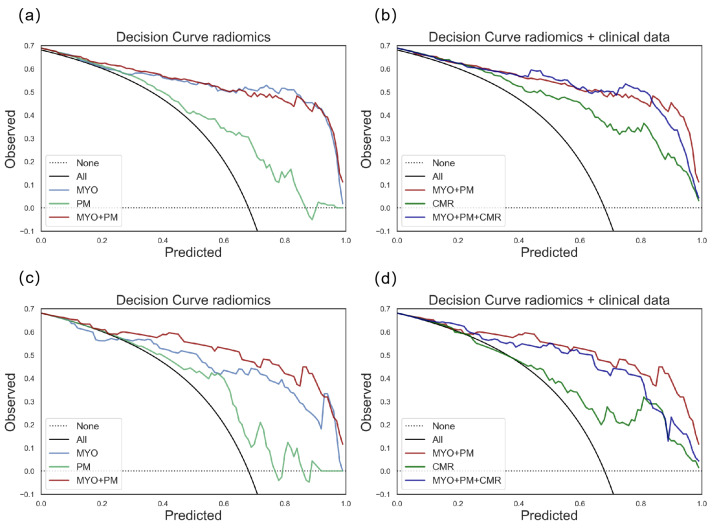
Decision curves for clinical usefulness evaluation. (**a**) shows the decision curves for the training dataset with the MYO, PM, and MYO+PM groups; (**b**) shows the decision curves for the training dataset with the radiomics and clinical data groups; (**c**) shows the decision curves for the testing dataset with the MYO, PM, and MYO+PM groups; (**d**) shows the decision curves for the testing dataset with the radiomics and clinical data groups.

**Figure 7 diagnostics-13-01544-f007:**
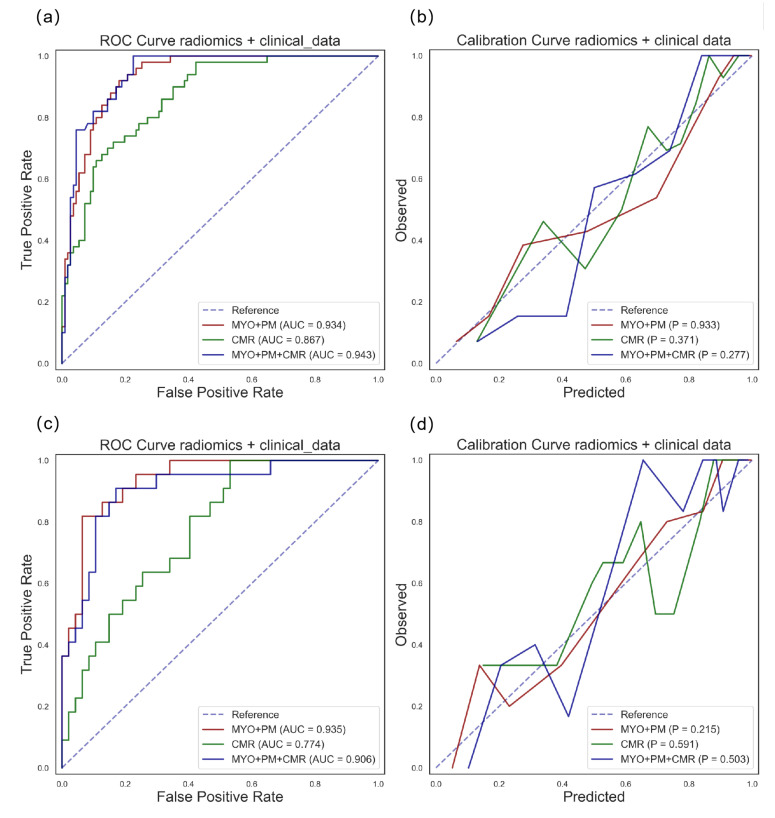
ROC curve and calibration curve for differentiation task with radiomics and clinical data using SVM model. Red, green, and blue lines represent radiomics, CMR parameters, and radiomics + CMR parameters groups, respectively. The AUC for the ROC curves and Hosmer–Lemeshow *p*-values are denoted in the figure legends. The first row shows the results on the training dataset: ROC curve (**a**) and calibration curve (**b**). The second row shows the results on the testing dataset: ROC curve (**c**) and calibration curve (**d**).

**Table 1 diagnostics-13-01544-t001:** Demographic and CMR characteristics.

Clinical Data Entries	Overall (N = 345)	LVH (N = 230)	HCM (N = 158)	HHD (N = 72)	HC (N = 115)	*p*-Value ^1^
**Demographic and Clinical Features**						
Age, year	51.1 ± 15.7	51.4 ± 15.3	51.6 ± 15.3	50.9 ± 15.4	50.6 ± 16.6	0.634
Male, n (%)	245 (71)	165 (72)	109 (69)	56 (78)	80 (70)	0.171
Weight, kg	71.5 ± 14.3	72.4 ± 14.0	70.2 ± 12.8	77.3 ± 15.4	69.8 ± 14.7	**0.001**
Height, cm	168.5 ± 9.0	168.3 ± 8.4	168.5 ± 8.3	170.0 ± 8.5	169.1 ± 9.9	l**0.048**
BMI, kg/m${}^{2}$	25.1 ± 3.9	25.4 ± 3.8	24.9 ± 3.5	26.6 ± 4.2	24.3 ± 3.8	l0.205
BSA, m${}^{2}$	1.79 ± 0.22	1.80 ± 0.21	1.77 ± 0.20	1.87 ± 0.23	1.77 ± 0.23	l**0.003**
**CMR Parameters**						
LVEF, %	64.1 ± 10.6	63.5 ± 11.9	67.5 ± 7.4	54.8 ± 15.0	65.4 ± 7.2	l **< 0.001**
LVEDV, mL	135.3 ± 40.7	140.2 ± 42.3	130.1 ± 29.6	162.5 ± 55.7	125.5 ± 35.5	l **< 0.001**
LVEDV index, mL/m${}^{2}$	75.2 ± 19.0	77.7 ± 20.3	73.8 ± 15.6	86.2 ± 26.2	70.2 ± 14.8	l**0.002**
LV mass, g	123.9 ± 53.2	144.3 ± 52.1	146.4 ± 54.4	139.7 ± 46.6	83.2 ± 24.0	l0.515
LV mass index, g/m${}^{2}$	68.8 ± 27.9	80.0 ± 27.3	82.7 ± 29.1	74.1 ± 21.9	46.3 ± 9.5	l**0.031**

Values are the mean ± standard deviation or number (percentage). ^1^
*p*-values were calculated between the HCM and HHD groups. *p*-values < 0.05 are shown in bold. BMI = body mass index; BSA = body surface area; CMR = cardiac magnetic resonance; HC = healthy control; HCM = hypertrophic cardiomyopathy; HHD = hypertensive heart disease; LV = left ventricular; LVEF = left ventricular ejection fraction.

**Table 2 diagnostics-13-01544-t002:** The 6 most-important features selected in the MYO and PM groups for HCM vs. HHD according to the Boruta method.

MYO (N = 6)	Relative Importance	PM (N = 6)	Relative Importance
**gradient GLCM correlation**	**37.5**	**original shape maximum 2D diameter slice**	**10.4**
**original shape sphericity**	**4.9**	**log-sigma-5-0-mm-3D first-order kurtosis**	**3.6**
**original shape elongation**	**3.5**	**original GLSZM ZoneEntropy**	**3.2**
wavelet-LHL GLCM Imc1	3.0	wavelet-HLL GLCM IMC2	3.0
log-sigma-5-0-mm-3D glszm ZoneEntropy	1.3	log-sigma-2-0-mm-3D GLCM correlation	2.9
wavelet-LHH GLCM MCC	1.2	gradient GLCM IDMN	2.4

The three most-important features in the MYO and PM groups were selected to construct the MYO+PM group. GLCM = gray-level co-occurrence matrix; GLSZM = gray-level size zone matrix; IDMN = inverse difference moment normalized; IMC = informational measure of correlation; MCC = maximal correlation coefficient; the detailed calculation method for each feature can be found on Pyradiomics’ official website.

**Table 3 diagnostics-13-01544-t003:** Performance of selected models on LVH detection and HCM vs. HHD differentiation with MYO, PM, and MYO+PM features.

Group	Feature Number	AUC ^1^	Accuracy (%)	Precision	Recall	F1 Score
**LVH detection task (SVM model)**						
MYO	6	**0.966** ^2^	90.4	0.903	**0.829**	0.853
PM	6	0.772 ^3^	68.3	0.676	0.486	0.507
MYO+PM	3 + 3	0.964	**91.3**	**0.913**	**0.829**	**0.866**
**HCM vs. HHD differentiation task (SVM model)**						
MYO	6	0.875 ^4^	82.6	0.831	0.773	0.739
PM	6	0.716 ^5^	73.9	0.811	0.182	0.308
MYO+PM	3 + 3	**0.935**	**87.0**	**0.871**	**0.818**	**0.800**

^1^ *p*-values were calculated comparing the MYO+PM group using the Delong test. ^2^ *p* = 0.924. ^3^
*p* < 0.001. ^4^
*p* = 0.040. ^5^
*p* = 0.002.

**Table 4 diagnostics-13-01544-t004:** SVM model accuracy of clinical-based features (including demographic and CMR features) and radiomics-based features on testing dataset.

Evaluation Metrics	CMR Parameters (LVEF + LVEDV Index + LV Mass Index)	Radiomics (MYO + PM)	Radiomics + CMR Parameters
AUC ^1^	0.774 ^2^	**0.935**	0.906 ^3^
Accuracy (%)	71.0	**87.0**	85.5
Precision	0.693	**0.871**	0.860
Recall	0.409	**0.818**	**0.818**
F1-score	0.474	**0.800**	0.783

^1^ *p*-values for the AUC were calculated comparing the radiomics group using the Delong test. ^2^ *p* = 0.002. ^3^
*p* = 0.117.

**Table 5 diagnostics-13-01544-t005:** Summary table of main findings in this research.

Task Name	Matrices	MYO	PM	MYO+PM	CMR	MYO+PM+CMR
LVH detection	AUC	**0.966**	0.772	0.964	0.908	0.965
	Accuracy	90.4	68.3	**91.3**	81.7	**91.3**
HCM vs. HHD differentiation	AUC	0.875	0.716	**0.935**	0.774	0.906
	Accuracy	82.6	73.9	**87.0**	71.0	85.5

Bold numbers indicate the highest AUC and accuracy with the selected feature group.

## Data Availability

No new data were created, and the data are unavailable due to privacy or ethical restrictions.

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
