# Peer review of "Papillary-Muscle-Derived Radiomic Features for Hypertrophic Cardiomyopathy versus Hypertensive Heart Disease Classification"

_diagnostics, 2023, doi:10.3390/diagnostics13091544_

Round 1

Reviewer 1 Report

In the presented study, the authors aimed to determine the value of radiomic features derived from papillary muscle (PM) in subjects with left ventricular hypertrophy (LVH) and healthy controls (HC). Also, they sought to develop models to detect LVH and differentiate hypertrophic cardiomyopathy (HCM) from hypertensive heart disease (HHD) via PM-derived features.

After analyzing 345 subjects undergoing cardiovascular magnetic resonance (CMR) examinations, they concluded that the proposed PM radiomics showed excellent performance in the detection of LVH and HCM relative to HHD differentiation and outperformed clinical data models. Moreover, they hypothesized that PM radiomics has great potential for the classification of other heart diseases and could facilitate clinical decision-making for clinicians.

The research is adequately conceived, the work is well written, the results are presented logically and in detail (textual, tabular, graphical). The topic is interesting for clinical practice. In this light, the presented study represents a small advance in the detection and differentiation of different types of LVH.

.

Author Response

Dear reviewer:

Thank you very much for your time and efforts on this manuscript.

We have read your comments and found it very encouraging for me and my colleagues.  Before submitting this manuscript, we performed one round of professional language editing service with an authorized organization  (The editing certificate is provided in attached file). Therefore, in this revised version, we double checked the full article and focused on the scientific English, format of figures and tables.  

Hoping this revised version could provide audiences with a more comfortable reading experience. More importantly, we hope the revised version could convey our study design, experiment details, results and insights on radiomics/ papillary muscle more explict and straight forward to all researchers.

Kind regards !

Qiming Liu

Shanghai Jiao Tong University

School of Medicine

Reviewer 2 Report

The manuscript  "Papillary muscle derived radiomic features for hypertrophic cardiomyopathy versus hypertensive heart disease classification"  by  Qiming Liu et al. it is well laid out and the pictures justify the research. I believe that a summary table is needed in the conclusions so that the understanding of the work done is faster.

Author Response

Dear reviewer:

Thank you very much for your time and efforts on this manuscript.

I have read your suggestion and I found it very helpful. In the revised version, we add a summary table in the conclusion part (right after the ‘4.1 Summary of main findings’ section). We also optimized the presentation form of ‘summary of main findings’ by using a more straight forward bulleted list.

Hoping this new table and bulleted list could help audiences understand and catch the main findings of this research work more direct and rapid!

Kind regards !

Qiming Liu

Shanghai Jiao Tong University

P.S. For your convenience, I paste the revised 'summary of main findings' in the attached file.

School of Medicine

Reviewer 3 Report

In this study authors aimed to determine the value of Papillary muscle derived radiomic features in subjects with LVH (Hypertensive heart disease and HCM) especially to differentiate between the two groups. It is difficult to determine VOI of Papillary muscle with certainity, would like to appreciate the authors using experienced cardiologist help in determining this. The study is well conducted, results are explained in detail and well written. The abstract is well written too.  Just recommend few minor changes:

Please clarify this exclusion sentence, i am unable to understand it: "subjects experienced activity with sufficient duration, intensity, and frequency to explain the abnormal LV wall thickness"? 

Expand the abbreviations in Figure 1.

I think gender word should be used cautiously. I think by gender authors meant biological sex (male or female). If that is the case, i would recommend changing gender to sex throughout the manuscript.

In table 1, would recommend entering p value as the last column.  

Thanks for considering the possibility of overfitting. And great job in adding clinical perspectives. 

Overall, the quality is english language is good, just few minor grammatical edits are needed. 

Author Response

Dear reviewer:

Thank you very much for your time and efforts spent on this manuscript.

I have read your suggestion and found it of great help.I summarized your suggestions, and list them with our reply below.

*1*: [clarify this exclusion sentence. "subjects experienced activity with sufficient duration, intensity, and frequency to explain the abnormal LV wall thickness"]

The purpose of this sentence is to declare that we have excluded left ventricular hypertrophy (LVH) caused by athlete’s heart. However, it’s my fault that I did not explain clearly. I checked some other articles and decided to use “LVH caused by athlete’s heart[1]” 

*2*:[Expand the abbreviations in Figure 1.]

I have listed the abbreviations in the Figure caption in the revised version. Also, I checked other figures, and added the abbreviations of machine learning algorithm (RF, SVM etc.) in the caption of Figure 3.

*3*:[ i would recommend changing gender to sex throughout the manuscript]

Thanks for your kind suggestion, I have changed ‘gender’ to ‘sex’ in the text/ figures and tables. I did not notice the difference between ‘gender’ and ‘sex’ before, and I would pay more attention to these words in the future.

*4*:[In table 1, would recommend entering p value as the last column.]

P-values are added to table 1 in the revised version. However, because the superior task in this research is to differentiation HCM and HHD. Therefore, we only provided the P-values between HCM and HHD groups for clarity.

Personally, I appreciate your kind and encouraging comments, which means a lot to a young researcher!

Kind regards!

Qiming Liu

Shanghai Jiao Tong University

School of Medicine

Reference:

[1] Prakken NH, Velthuis BK, Teske AJ, Mosterd A, Mali WP, Cramer MJ. Cardiac MRI reference values for athletes and nonathletes corrected for body surface area, training hours/week and sex. Eur J Cardiovasc Prev Rehabil. 2010 Apr;17(2):198-203.